# Optimal free descriptions of many-body theories

Christopher J. Turner[1], Konstantinos Meichanetzidis[1], Zlatko Papić[1] & Jiannis K. Pachos[1]

Interacting bosons or fermions give rise to some of the most fascinating phases of matter, including high-temperature superconductivity, the fractional quantum Hall effect, quantum spin liquids and Mott insulators. Although these systems are promising for technological applications, they also present conceptual challenges, as they require approaches beyond mean-field and perturbation theory. Here we develop a general framework for identifying the free theory that is closest to a given interacting model in terms of their ground-state correlations. Moreover, we quantify the distance between them using the entanglement spectrum. When this interaction distance is small, the optimal free theory provides an effective description of the low-energy physics of the interacting model. Our construction of the optimal free model is non-perturbative in nature; thus, it offers a theoretical framework for investigating strongly correlated systems.

[1] School of Physics and Astronomy, University of Leeds, Woodhouse Lane, Leeds, Yorkshire LS2 9JT, UK. Correspondence and requests for materials should be addressed to C.J.T. (email: pycjt@leeds.ac.uk) or to K.M. (email: mmkm@leeds.ac.uk).

Many-body physics of non-interacting systems reduces to the description of a single particle and thus is well understood. In contrast, the understanding of interacting systems remains one of the major open problems of both condensed matter and high-energy physics. Complete analytical solutions are possible in the so-called integrable systems[1], which are not robust to perturbations. More generally, one relies on mean-field approaches, density functional theory or perturbation theory to expand around known solvable instances. Such methods can be employed when correlations are weak or interactions induce small corrections to the original state. Many interesting phenomena, however, have non-perturbative origin, such as superconductivity or the fractional quantum Hall effect. Although important insights about such systems have been obtained using variational ansätze[2–5], this approach requires non-trivial physical intuition about the nature of the emerging free quasiparticles. A question of paramount importance arises: is it possible to directly identify the free effective model that is most similar to a given interacting one?

Here we introduce the concept of the interaction distance, $D_{\mathcal{F}}$, which quantifies the effect of interactions on the ground state of a many-body system. At the same time, we identify the optimal free theory closest to the given interacting model. Our approach employs quantum information inspired techniques to study the correlations of a system witnessed in its entanglement spectrum and to build a general diagnostic tool of interactions. Typically, we find $D_{\mathcal{F}}$ to be small when mean-field theory is applicable, whereas non-trivial behaviour emerges near critical regions. Using the example of the quantum Ising model, we demonstrate that the interaction distance can be calculated efficiently. We envision that finding the optimal free description of interacting systems can help formulate suitable variational ansätze in a variety of areas, ranging from condensed matter to high energy physics. Alternatively, it could be used to develop efficient numerical simulations of interacting systems that scale favourably with the system size.

## Results

**Overview**. The interaction distance $D_{\mathcal{F}}$, introduced below, quantifies the distance between the reduced density matrix of a many-body state and the closest free-particle reduced density matrix. We demonstrate that $D_{\mathcal{F}}$ can be calculated efficiently from the entanglement spectrum[6] and when this distance is small ($D_{\mathcal{F}} \ll 1$) leads to the optimal free model that best describes the low-energy properties of the interacting system. Near critical regions, $D_{\mathcal{F}}$ behaves non-trivially and its finite-size scaling can be related to the properties of the model under renormalization group flow. As an example, we apply our method to the one-dimensional (1D) quantum Ising model in the presence of transverse and longitudinal fields. We demonstrate that this model has $D_{\mathcal{F}} \approx 0$ across the whole phase diagram and we identify its optimal free description. Finally, we present a particular model with non-zero interaction distance in the thermodynamic limit, thus demonstrating its intrinsic interacting nature.

**Interaction distance and optimal free model**. Consider an arbitrary many-body system prepared in its ground state, $|\Psi\rangle$. For simplicity, here we consider fermionic systems defined on a lattice, although our approach can be generalized to other systems, as we discuss below. Partitioning the system in two regions, $A$ and its complement $B$, defines the reduced density matrix $\rho = \mathrm{tr}_B |\Psi\rangle\langle\Psi|$. The entanglement Hamiltonian $H_{\mathrm{E}} = -\ln \rho$ has eigenvalues $\{E\}$, known as the entanglement spectrum[6]. The

entanglement spectrum captures the correlations in the ground state. Moreover, the universal properties of the actual Hamiltonian of the interacting system are reflected in $H_{\mathrm{E}}$[6–9]. For this reason, we diagnose the effect of interactions and identify the optimal free model exclusively through ground-state correlations.

We introduce the interaction distance between the interacting $\rho$ and the free $\sigma$ reduced density matrices

$$D_{\mathcal{F}}(\rho) = \min_{\sigma \in \mathcal{F}} D(\rho, \sigma), \qquad (1)$$

where $D(\rho, \sigma) = \frac{1}{2} \mathrm{tr} \sqrt{(\rho - \sigma)^2}$ is the trace distance and $\mathcal{F}$ is the manifold of all free fermion states. It is worth noting that unlike previous works[10–16], the manifold $\mathcal{F}$ contains all Gaussian states in any set of fermionic quasiparticle operators $\{c\}$. The trace distance has a physical interpretation in terms of distinguishability between $\rho$ and $\sigma$ when measuring observables[17]. Alternative state-distance measures can equally well be employed[18]. The quantity $D_{\mathcal{F}}$ has a geometric interpretation as the distance of the density matrix $\rho$ from $\mathcal{F}$.

To compute $\mathcal{D}_{\mathcal{F}}$, we need to perform the minimization in equation (1). We can show that this problem can be reduced to varying only the spectrum of $\sigma$. Consider the basis where $\rho$ is diagonal and its entanglement spectrum $\{E\}$ is available. A general $\sigma \in \mathcal{F}$ need not be diagonal in the same basis as $\rho$. However, the trace distance between $\rho$ and $\sigma$ is minimized when $\sigma$ commutes with $\rho$, that is, it is simultaneously diagonal with $\rho$ and their eigenvalue spectra are rank ordered[19]. Indeed, if there existed a $\sigma$ which minimized $D_{\mathcal{F}}$ but did not commute with $\rho$, then that minimum would not be a global one. As a consequence, the eigenstates of the optimal model $\sigma$ are the same as the eigenstates of $\rho$. Thus, the minimization for obtaining $D_{\mathcal{F}}$ now involves a variation with respect to the eigenvalues of $\sigma$ that can be given in terms of its entanglement spectrum.

Having to minimize only with respect to the spectrum of $\sigma$ is a significant simplification of the optimization problem. A further exponential simplification is possible if we consider the structure of its entanglement spectrum. Consider $N$ single-particle entanglement energies $\{\epsilon\}$ with allowed occupations $\{n_i, i = 1,\ldots, N; n_i = 0, 1\}$ corresponding to the independent modes $\{c\}$ obeying Fermi-Dirac statistics. The full entanglement spectrum $\{E^{\mathrm{f}}\}$ of $\sigma$ contains exponentially many levels, $2^N$, as a function of subsystem size $N$. However, a special property of free systems is that due to Wick's theorem their entanglement spectrum can be built from a set of single-particle entanglement energies $\epsilon_i$ according to[20]

$$E_k^{\mathrm{f}}(\{\epsilon\}) = E_0 + \sum_{i=1}^{N} n_i(k)\, \epsilon_i, \qquad (2)$$

where $E_0$ is a normalization constant. The index $k$ runs over the many-body spectrum, and for each $k$ there are associated occupations $n_i(k) \in \{0,1\}$. Hence, the interaction distance, $D_{\mathcal{F}}$, can be cast as a minimization with respect to the $N$-many single-particle energies

$$D_{\mathcal{F}}(\rho) = \min_{\{\epsilon\}} \frac{1}{2} \sum_k \left| \mathrm{e}^{-E_k} - \mathrm{e}^{-E_k^{\mathrm{f}}(\{\epsilon\})} \right|. \qquad (3)$$

As $N$ increases at most linearly with system size, expression (3) provides the means to efficiently compute the interaction distance and obtain the optimal free model of any state of an interacting theory whenever its entanglement spectrum $\{E\}$ is accessible.

According to (3) the interaction distance is zero when the entanglement spectrum of $\rho$ satisfies the combinatorial structure given in (2) for certain single-particle energies, $\{\epsilon\}$. This generalizes our concept of free models. In general, no $\epsilon$'s exist that satisfy all these constraints as their number grows exponentially with the system size, while the number of $\epsilon$'s grows

only linearly. Owing to the properties of the trace distance[18] we have that the interaction distance takes values $D_\mathcal{F} \in [0,1]$. The condition $D_\mathcal{F} = 0$ corresponds to a system that can be exactly described by the free fermions $\{c\}$, while $D_\mathcal{F} = 1$ is the maximal distance a state can be from a free description. When $D_\mathcal{F}$ is approximately zero, then the deviation in expectation values or correlation functions between $\rho$ and the approximation $\sigma$ is bounded. In particular, the interaction distance is sensitive only to deviations in the low lying entanglement spectrum, which dominate the expectation values of physical observables.

At this point we might wonder what significance the bipartition holds in definition (1) of the interaction distance. From a quantum information point of view, the partial trace serves as a quantum channel through which we view the ground state. This channel is Gaussian if it maps a Gaussian state to another Gaussian state[21]. The combinatorial structure of (2) describes the eigenvalue spectrum of any Gaussian density matrix. In matching the spectrum we are testing compatibility between canonical operators in which the original state is a Gaussian state and the quantum channel is Gaussian, and our ability to describe the state in terms of modes separable to $A$ and $B$. If there exist such modes then $D_\mathcal{F}$ is zero.

A useful insight in the form of the modes $\{c\}$ is given by a constructive derivation of (3) from (1). The canonical algebra of $\{c\}$ is invariant under arbitrary unitary transformations on $\sigma$. Hence, these transformations keep $\sigma$ in $\mathcal{F}$ and $\mathcal{F}$ contains all of its unitary orbits. Optimizing for the minimum $D_\mathcal{F}$ involves the following steps. First, $\mathcal{F}$ is decomposed into equivalence classes which are the mentioned unitary orbits. Within each class, the trace distance is minimized by a certain representative $\sigma$ with which $\rho$ commutes[19]. Then, $D_\mathcal{F}$ is obtained by taking the minimum over representatives of each class. As within these unitary orbits the trace distance is minimized when $\sigma$ and $\rho$ are simultaneously diagonal, the free modes $\{c\}$ are given as Schmidt vectors corresponding to the single-particle entanglement levels $\{\epsilon\}$ for which we have optimized. Note that the unitarily transformed modes are, in general, a nonlinear combination of the original modes, though they still describe a free model. From the optimal state $\sigma$, we can identify an effective free physical Hamiltonian in terms of the emergent quasiparticles $\{c\}$ and we refer to this as the optimal free model. It can be found via a non-unique procedure[22,23] which utilizes the two-point correlations of $\{c\}$ with respect to $\sigma$.

**General optimal free description.** To determine the value of $D_\mathcal{F}$ we have chosen in (2) the statistics of the free quasiparticles to be fermionic. Generalizing further, we could allow for an optimal free description with respect to other quasiparticle statistics than the constituent fermions of the original interacting system. However, systems that comprize quasiparticles with different mutual statistics have in general Hilbert spaces of unequal dimension. This can be directly resolved in the following way. The entanglement spectrum of a pure state $|\psi\rangle$ can be found from the Schmidt decomposition, $|\psi\rangle \cong V_A \Sigma V_B^\dagger$, where the state has been reinterpreted as a map between the Hilbert spaces of the $A$ and $B$ parts. The isometries $V_A$ and $V_B$ map from $A$ to $B$ through an intermediate space $\mathcal{H}_E$, the entanglement Hilbert space on which the matrix of singular values $\Sigma$ and the related entanglement Hamiltonian, $H_E$, are defined. One can increase the dimension of $\mathcal{H}_E$ by the addition of zero singular values to $\Sigma$ and the addition of linearly independent columns to $V_A$ and $V_B$ so that they remain full-rank, without changing the correlations in $|\psi\rangle$. If the dimension of $\mathcal{H}_E$ is made to exceed the Hilbert space dimension of either subsystem then the isometries serve as

projectors removing non-physical states with high entanglement energy. This fact allows us to compare interacting systems with optimal free models that may have different Hilbert space dimension.

**Efficiency of computing $\mathbf{D}_\mathcal{F}$.** In the following we shall demonstrate with two examples that the interaction distance, $D_\mathcal{F}$, is a versatile quantity that can be evaluated numerically or analytically. Importantly, the calculation of $D_\mathcal{F}$ requires only the knowledge of the ground state; thus, it is more efficient than other possible measures, for example, based on directly comparing the structure of actual energy spectra.

When $\rho$ is the reduced density matrix of a gapped 1D system, then $D_\mathcal{F}$ can be numerically determined efficiently with $L$, the linear size of the system. Gapped 1D states have area-law entanglement[24], which bounds the number of significant levels in the entanglement spectrum in the thermodynamic limit. Below we use the well-known density matrix renormalization group algorithm (DMRG)[25] to efficiently obtain ground states of finite 1D systems. Moreover, in this case the low lying $\{\epsilon\}$ in (2), and as a consequence the state $\sigma$, will converge exponentially fast with $L$. Thus, the minimization procedure in (3) is also efficient (see Methods), as it need only involve a finite number of significant levels determined by the correlation length, even in the thermodynamic limit.

For critical 1D systems, logarithmic corrections to the area law are possible, which leads to the polynomial complexity in determining $\rho$. For critical 1D states, a multi-scale renormalization ansatz[26] can be implemented to obtain the entanglement spectrum. If the system is gapless, the number of significant entanglement levels will increase but only polynomially with system size $L$; hence, the optimization procedure for determining the single-particle energies $\epsilon$ remains efficient.

Finally, in higher-dimensional systems, our method is reliant on the efficiency of the current methods in the literature for computing the entanglement spectrum of the ground state. For two-dimensional (2D) systems, one can use iterative methods such as the Lanczos algorithm to access only the ground state in the exact diagonalization framework. Furthermore, Monte-Carlo algorithms[27] and 2D tensor networks[28,29] can be used in a variety of systems to variationally approximate the ground state. Then the typical runtime complexity of $D_\mathcal{F}$ for an entanglement spectrum from a disk partition is polynomial in the correlation length and thus efficient.

**Finite-size scaling of $\mathbf{D}_\mathcal{F}$.** The quantity $D_\mathcal{F}$, through its dependence on the entanglement spectrum, inherits the information about both short- and long-wavelength properties of the system. As pointed out by Li and Haldane[6], the entanglement spectrum of a gapped phase exhibits a generic separation into the universal long-wavelength part and a non-universal short-distance part, the two being separated by the entanglement gap[6]. Assuming that the linear size of the system's quasiparticles, $\ell$, is much smaller than the linear size of the partition $A$, the long-wavelength information corresponds to correlated quasiparticle excitations across the entanglement partition, whereas the short-distance physics is associated with internal structure of the quasiparticles. The non-universal part is thus a boundary effect which is insensitive to variation in the subsystem size. In the thermodynamic limit, the non-universal part is exponentially suppressed in a gapped phase[6], as seen from (3), and $D_\mathcal{F}$ then predominantly describes the universal properties of the system.

At critical points where the quasiparticles remain well defined, that is, $\ell$ stays finite, a large but finite system of linear size $L \gg \ell$

chops off some of the correlations between the quasiparticles. We surmise that the finite-size scaling of $D_\mathcal{F}$ near such critical regions follows the ansatz

$$D_\mathcal{F} \approx \left(L^{-1} + \theta\right)^\zeta f\left((g - g_c)L^{1/v}\right), \qquad (4)$$

where $f$ is an undetermined function, and $\zeta$ and $v$ are the critical exponents. The constant $\theta \geq 0$, which vanishes in the standard power-law scaling ansatz[30], accommodates the fact that the interaction distance is bounded from above, $D_\mathcal{F} \in [0, 1]$. A simple scaling analysis (see Methods) shows that $v$ is the correlation length exponent, whereas $\zeta$ determines the effect of interactions in the renormalization group sense. For example, when interactions are relevant, $D_\mathcal{F}$ should remain non-zero as $L$ increases, which dictates $\zeta \leq 0$. On the other hand, when interactions are irrelevant, it is expected that $D_\mathcal{F}$ decreases with $L$ near the critical regions, in which case $\zeta > 0$. However, it is noteworthy that it is possible for interactions to be irrelevant and still yield finite $D_\mathcal{F}$ in the thermodynamic limit. This is because, $D_\mathcal{F}$ may be sensitive to non-universal (short distance) properties of the system, which can give a residual non-zero contribution parametrized by $\theta$.

**Application to Ising model.** For concreteness we consider the example of the 1D ferromagnetic (FM) and antiferromagnetic (AFM) Ising model in both transverse, $h_z$, and longitudinal, $h_x$, fields (see ref. 31 for a recent review). By using exact diagonalization to determine the entanglement spectrum, we compute $D_\mathcal{F}$ across the phase diagram and examine its scaling around criticality and its convergence in the thermodynamic limit. Finally, we find optimal free-fermion models for each point $(h_z, h_x)$ of the phase space.

The interacting Hamiltonians are given by

$$H_\pm = -\sum_{j=1}^{L} \left( \pm \sigma_j^x \sigma_{j+1}^x + h_z \sigma_j^z + h_x \sigma_j^x \right), \qquad (5)$$

where $H_+$ stands for FM and $H_-$ for AFM with periodic boundary conditions. In the presence of only transverse field ($h_x = 0$), model (5) maps to free fermions via the Jordan-Wigner transformation. A non-zero longitudinal field, $h_x$, introduces non-local interactions between fermions. A quantum critical point at $h_z = 1$ separates an ordered and a disordered phase of the free system which are related by a self-duality[32]. The FM model has a single critical point at ($h_z = 1$, $h_x = 0$), whereas the AFM model has a critical line connecting ($h_z = 1$, $h_x = 0$) with the classical point ($h_z = 0$, $h_x = 2$)[33].

Minimizing the interaction distance over the phase diagram we find that $D_\mathcal{F}$ decays with $L$ away from critical regions as shown in Fig. 1, with the variational parameters $\{\epsilon\}$ converging exponentially (see Methods). Thus, the model can be faithfully described by a free theory in these regions of the phase diagram. The exceptions only occur infinitesimally close to the FM critical point and at the AFM classical critical point. This is remarkable because these models are non-integrable and a priori have strong quantum fluctuations due to all energy scales being comparable in magnitude.

Strong correlations build up near criticality where the effect of interactions is most significant and $D_\mathcal{F}$ can take higher values. These regions, however, shrink around criticality as $L$ increases. To examine this we consider the finite-size scaling of $D_\mathcal{F}$ using ansatz (4), as shown in Fig. 2, along the paths $h_z = 1$ (FM) and $h_x = h_z$ (AFM) shown in Fig. 1. It is noteworthy that, despite being near criticality, the values of $\mathcal{D}_\mathcal{F}$ remain small ($\mathcal{D}_\mathcal{F} \ll 1$) for both models; thus, we set $\theta = 0$ in the ansatz (4). We obtain critical exponents $\zeta_{FM} \approx -1.4$ and $\zeta_{AFM} \approx 1.11$, showing that interactions have a dramatically different effect in the two models. In the FM case the interactions are a singular perturbation to the critical point, whereas in the AFM case their effect diminishes as $L$ increases. This behaviour is consistent with the shrinking of the significant high $D_\mathcal{F}$ regions around criticality. Furthermore, the critical exponent $v_{FM} \approx 0.533$, is approximately equal to that of the correlation length $v_{FM}^\xi = 8/15$ (ref. 34). Similarly, $v_{AFM} \approx 1.252$ which is within 20% accuracy to the correlation length critical exponent for the same cut $v_{AFM}^\xi \approx 1.052$, which we obtain numerically. We account for the deviation in $v_{AFM}$ by small system sizes and the fact that we are not perturbing with an operator that has a well-defined scaling dimension. The critical scaling behaviour of $D_\mathcal{F}$ is independently verified by employing the variational DMRG method rather than exact diagonalization, which can efficiently probe significantly larger system sizes (see Methods).

Finally, we are in position to identify the optimal free model that describes the interacting system given by an instance of (5). In particular, we identify the free Ising model $H_\pm(h_z^f, 0)$ with transverse field $h_z^f$, whose ground state's entanglement spectrum matches $\sigma$'s obtained from (1) for each point $(h_z, h_x)$. This is simply obtained by minimizing $D(\sigma(h_z, h_x), \sigma^f(h_z^f, 0))$ over $h_z^f$. As a result we observe that in the FM case, adding infinitesimal interactions to the free Ising model with $h_z < 1$ maps the model to a free Ising with $h_z > 1$ in a discontinuous way, as shown in Fig. 3a. When $h_z > 1$, the introduction of interactions maps the model to a neighbouring free model continuously. In the AFM case, the interactions are irrelevant. Indeed, Fig. 3b shows that the whole phase diagram maps trivially to the free model even very near criticality.

The distance $\min_{h_z^f} D(\sigma, \sigma^f)$ is shown by the colour scale in Fig. 3. We see that away from criticality $\sigma$ can be mapped to the free Ising model with a high fidelity. In the thermodynamic limit we expect the critical line of the AFM to also be identified with a free Ising model, because $\min_{h_z^f} D(\sigma, \sigma^f)$ decreases with $L$ and the conformal field theory, which describes the point ($h_z = 1$, $h_x = 0$), also governs the entire critical line[31]. This analysis reveals that the optimal free model is local and fermionic throughout the phase diagrams, except at the critical point of the FM model and the classical critical point of the AFM model whose ground state is macroscopically degenerate.

**Maximally interacting model.** In the above analysis, we found for the Ising model that the interaction distance vanishes in the thermodynamic limit in almost the entire phase diagram. For that model we identified the optimal free fermion model that

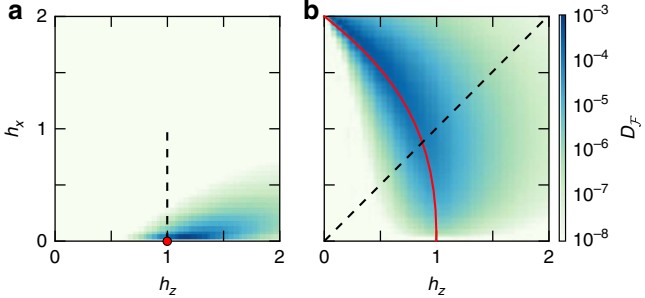

**Figure 1 | Interaction distance across the phase diagram of the quantum Ising model.** (**a**) The FM and (**b**) the AFM model at system size $L = 16$ with periodic boundary conditions. The interaction distance (log scale) takes non-negligible values only in the vicinity of the critical point and critical line which are sketched in red. The scaling behaviour of $D_\mathcal{F}$ along the dashed lines is given in Fig. 2.

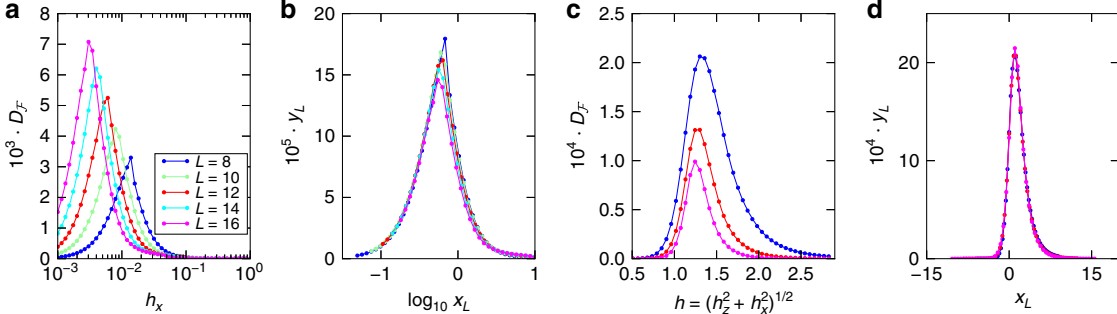

**Figure 2 | Finite-size scaling of the interaction distance for the Ising model.** $D_{\mathcal{F}}$ for a number of systems sizes of (**a**) the FM model and (**c**) the AFM model along the dashed lines given in Fig. 1. We obtain the critical exponents $\nu_{FM} \approx 0.533$, $\zeta_{FM} \approx -1.4$ and $\nu_{AFM} \approx 1.252$, $\zeta_{AFM} \approx 1.11$ by fitting to (4). (**b**,**d**) show the scaling collapse of $D_{\mathcal{F}}$, where $x_L = (g - g_c)L^{1/\nu}$ and $y_L = D_{\mathcal{F}}L^{\zeta}$. This demonstrates the applicability of the scaling ansatz (4).

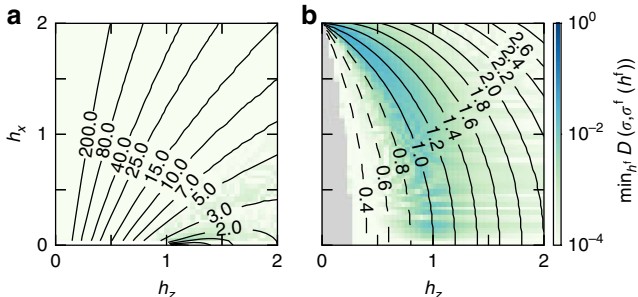

**Figure 3 | Mapping the optimal free model to a transverse-field Ising model.** (**a**) The FM model and (**b**) the AFM model both at system size $L = 16$. Contours indicate the transverse-field $h_z^f$, dashed lines for those in the symmetry broken phase and solid otherwise (including the critical value $h_z^f = 1$). The background colour plot gives the distance $\min_{h_z^f} D(\sigma, \sigma^f)$ (log scale) signifying the success of the mapping. In this way the interacting system is given a description in terms of a free-fermion model. The region near the $h_z = 0$ classical axis of the AFM is removed (grey region) because our calculations do not resolve all symmetries of the system.

effectively describes it throughout the phase space. We now present a truly interacting model that cannot be approximated by free fermions, that is, has a non-zero $D_{\mathcal{F}}$ in the thermodynamic limit, even away from criticality. In our analytical approach, we first construct the entanglement spectrum that gives a non-zero $D_{\mathcal{F}}$ with respect to free fermions. Then, we identify the physical system and emergent quasiparticles that correspond to the derived entanglement spectrum.

We consider the simple case where the entanglement Hamiltonian comprises two fermionic modes. We aim to maximize the interaction distance with respect to the entanglement spectrum $\{E_{max}\}$ which consists of four levels. We perform a maximization of the interaction distance from free-fermion entanglement spectra generated by two single-particle energies, $\epsilon_1, \epsilon_2$. It can be analytically proven (see Supplementary Note 2) that the maximum interaction distance in this case is

$$D_{\mathcal{F}}(\rho_{max}) = \max_{\{\lambda\}} D_{\mathcal{F}}(\rho) = \frac{1}{6}, \tag{6}$$

where the maximization is performed with respect to the eigenvalues $\{\lambda\}$ of $\rho$. The spectrum of the reduced state $\rho_{max}$ that maximizes $D_{\mathcal{F}}(\rho)$ is the maximally mixed rank-3 spectrum $\{\lambda_{max}\} = \{\frac{1}{3}, \frac{1}{3}, \frac{1}{3}, 0\}$, with entanglement spectrum $\{E_{max}\} = \{\ln 3, \ln 3, \ln 3, \infty\}$.

We would now like to find the parent physical system which exhibits such correlations in the ground state so that it saturates the maximum of $\mathcal{D}_{\mathcal{F}}$. As the entanglement spectrum $\{E_{max}\}$ has a

three-fold degeneracy, it is natural to consider models which support fractionalized excitations. In particular, a $\mathbb{Z}_3$ quantum clock model effectively describes the edge physics of a 2D topological phase[35] and can in principle be realized in the laboratory[35–39]. The $\mathbb{Z}_3$ Hamiltonian in the topological phase at its renormalization fixed point is

$$H_{\mathbb{Z}_3} = -\sum_j \tau_j^\dagger \tau_{j+1} + \text{h.c.}, \tag{7}$$

where $\tau_j$ are non-Hermitian clock operators which commute between sites and can be represented locally as $\tau_j = \text{diag}(1, e^{i2\pi/3}, e^{-i2\pi/3})$ satisfying $\tau_j^3 = \mathbb{1}$.

The ground state of (7) hosts topologically protected parafermionic zero-modes exponentially localized on open boundaries[40]. These correspond to three-fold degeneracy in the entanglement spectrum $\{E\} = \{\ln 3, \ln 3, \ln 3\}$ from maximally entangled pairs of parafermions across each virtual boundary. We have seen in a previous section that we can increase the dimension of the entanglement Hilbert space $\mathcal{H}_E$ by introducing completely uncorrelated states that correspond to infinite entanglement energy. Hence, the entanglement spectrum of the $\mathbb{Z}_3$ model reproduces the aforementioned $\{E_{max}\}$. As a result we have readily identified a truly interacting model that gives $D_{\mathcal{F}}(\rho_{max}) = 1/6$ with respect to free fermions in the thermodynamic limit. Importantly, we have managed to analytically identify the ground state of the corresponding interacting model by considering only the structure of its correlations.

## Discussion

In this study we have identified, for a given interacting theory, the optimal free state which is closest to its ground state and introduced the interaction distance between them. Quantifying interactions in such generic terms gives a fresh perspective into the physics of interacting systems. Indeed, our approach extends beyond mean-field theory, which is valid for weak correlations and spatial dimensions above the upper-critical dimension, or perturbation theory which requires weak couplings. Optimizing with respect to the correlations present in the ground state captures faithfully the low energy physics of the interacting system reproducing the observables with bounded error.

We have numerically determined the interaction distance for the 1D Ising model in the presence of transverse and longitudinal fields. We used both exact diagonalization as well as DMRG, thus demonstrating the efficiency in determining $D_{\mathcal{F}}$ for large 1D gapped systems using standard techniques. Further, our diagnostic $D_{\mathcal{F}}$ shows that the ground state of this interacting model in its gapped phases is well described by a free state, requiring exponentially less information to represent than the

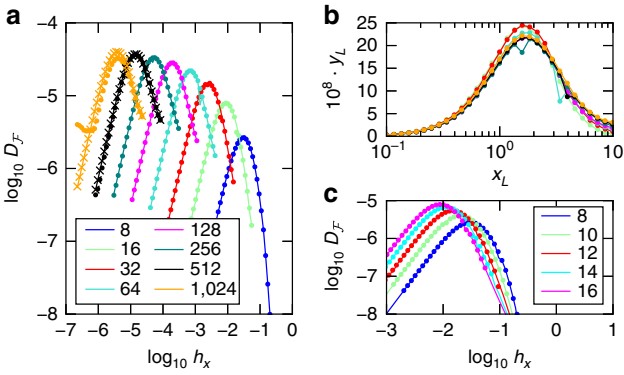

**Figure 4 | Interaction distance $D_{\mathcal{F}}$ for the FM Ising model calculated from the density matrix renormalization group.** (**a**) DMRG results retaining $\chi$ states for $\chi = 32$ (dots), 64 (lines) and 128 (crosses) states. The result is already converged for $\chi = 64$ for the biggest system size $L = 1{,}024$. (**b**) Scaling collapse according to ansatz (4) with the critical exponents $v \approx 0.533$, $\zeta \approx -1.4$ and saturation parameter $\theta \approx 0.025$. The collapsed variables $x_L = (g - g_c)L^{1/v}$ and $y_L = D_{\mathcal{F}}(1/L + \theta)^{-\zeta}$. (**c**) Comparison between $D_{\mathcal{F}}$ calculated for entanglement spectra provided via DMRG (points), for $\chi = 16$ entanglement levels, and exact diagonalization (lines) for the same system. The results for these system sizes accessible by exact diagonalization agree. All results in this figure are calculated for open boundary conditions where DMRG is most natural.

exact ground state. We expect that this could be generalized to an efficient algorithm for finding a representative (nearly) free state, similar to the DMRG method which constructs a low Schmidt rank approximation to gapped 1D ground states. Alternatively, our method can be combined with analytical wave functions, as in the case of the Bethe ansatz[41] or the trial wave functions in the quantum Hall effect[5,42,43].

We have also verified that there exist quantum states for which $D_{\mathcal{F}} \neq 0$, such as the $\mathbb{Z}_3$ quantum clock model. Other possible candidates are systems that give rise to exotic phases such as high-$T_c$ superconductors and states with intrinsic topological order, where interactions play a crucial role. This could establish $D_{\mathcal{F}}$ as an interaction order parameter identifying truly interacting systems, in terms of fermions or bosons, from nearly free ones such as the Ising model. Furthermore, as we have shown for the Ising Model and the $\mathbb{Z}_3$ model, it is possible to directly identify a parent Hamiltonian for the optimal free state.

In introducing our interaction distance, we have generalized the meaning of freedom in many-body quantum states by choosing the mutual statistics of the free modes we are optimizing over. Varying the statistics of the free modes used in our optimization corresponds to changing the free manifold $\mathcal{F}$ from which the interaction distance is measured. Our construction can be immediately generalized to soft-core bosons by promoting the occupation $m$ of each mode in equation (2) to a variational parameter taking values $1 \leq m < \infty$. Taking this further, the single-body levels themselves can become occupation dependent. Such a modification could accommodate fractionalized excitations in strongly-correlated states[44]. Another interesting generalization would be to introduce the notion of entanglement temperature[45], which shifts the sensitivity of the measure to other parts of the entanglement spectrum, reflecting entanglement on different length scales. Finally, we mention that methods for measuring entanglement spectra in optical lattices have recently been proposed[46]. Hence, the interaction distance can be determined for exotic states realized in cold atom systems.

## Methods

**Optimization.** The optimization to find $\sigma$ and $D_{\mathcal{F}}$ in (3) is performed by a Monte Carlo basin-hopping strategy[47] using the Nelder–Mead simplex algorithm for local minimization within basins of the cost function $D_{\mathcal{F}}$. This global strategy was selected to counteract an observed tendency for local methods to get trapped in local minima. The initial guess for this search is found as follows. The normalization constant $E_0$ is the lowest entanglement energy of the input entanglement spectrum $\{E\}$. We iteratively construct an approximate set of single-particle entanglement energies starting from an empty set. First, we take the lowest remaining level in the spectrum and subtract $E_0$ to produce a new single-particle level $\epsilon_k$. Then we remove the many-body levels, which are closest to the new combinatorial levels generated according to (2) by the additional single-particle level. This process is repeated until the input spectrum is exhausted. We can also introduce a truncation of the entanglement spectrum cutting off high entanglement energies, making the construction of the initial guess terminate faster. The minimization of $D(\sigma, \sigma^f)$ to identify the optimal free model for the Ising Hamiltonian (5) is calculated using a local Nelder–Mead method.

**Finite-size scaling.** We perform the finite-size scaling according to an ansatz (4). The parameters of the collapse were estimated using the method of ref. 48. From the scaling ansatz (4) and for a trial set of scaling parameters $g_c$, $v$ and $\zeta$, the scaled values $x_L = (g - g_c)L^{1/v}$ and $y_L = D_{\mathcal{F}}(1/L + \theta)^{-\zeta}$ are calculated from each unscaled data point $(g, D_{\mathcal{F}})$. From this collection of scaled data points $(x_L, y_L)$ across all $L$, we implicitly define a so-called master curve that best represents them. This curve $y(x)$ is defined around a point $x$ as the linear regression calculated by taking the scaled data points immediately left and right of $x$ for each system size $L$. We characterize the deviation of the scaled points $(x_L, y_L)$ from the master curve $y(x_L)$ using the $\chi^2$ statistic. This measure is used as the cost function for an optimization problem over the scaling parameters $g_c$, $v$, $\zeta$ and $\theta$, which can be solved using the same techniques as the previous problems.

In Fig. 4a, we show $D_{\mathcal{F}}$ calculated from the entanglement spectrum using DMRG[25], which extends our results from Fig. 2 in the FM case to larger system sizes inaccessible to exact diagonalization. In Fig. 4b, we show the scaling collapse. We obtain critical exponents $v$, $\zeta$, which are consistent with those found with exact diagonalization and $v$ is consistent with known results of CFT[34]. There are two differences compared with results in Fig. 2. First, with our larger sizes we become sensitive to the upper bound of $D_{\mathcal{F}}$, which gives us $\theta \approx 0.025$. The fact that $\theta$ is non-zero is readily apparent in the saturation behaviour visible in the unscaled data and is demanded for consistency with an upper bound. Second, in accordance with common practice, we perform DMRG using open boundary conditions, which changes the non-universal function $f$ in the scaling ansatz (4).

To verify our DMRG results we compare $D_{\mathcal{F}}$ obtained by DMRG and exact diagonalization with open boundary conditions. We confirm that they are in excellent agreement, as shown in Fig. 4c. Between iterations in DMRG, the ground state is approximated by retaining a reduced number of entanglement states. They correspond to the greatest Schmidt weights. The interaction distance is insensitive to these approximations inherent in DMRG, because the induced error is comparable to the truncation error which is typically kept to be $10^{-14}$. We demonstrate that our DMRG calculations are converged in the number of retained states $\chi$ in Fig. 4a. Close to criticality, due to the logarithmic growth in entanglement, we need to retain more states for larger $L$. The result is already converged for $\chi = 64$ at the largest $L$ studied, which justifies our approximation in retaining this number of states in the finite-size scaling. It is worth noting that the single-particle entanglement energies converge exponentially away from criticality exponentially (as discussed in Supplementary Note 1).

**Data availability.** All data presented in this work are available from the authors upon request. Statement of compliance with EPSRC policy framework on research data: this publication is theoretical work that does not require supporting research data.

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

## Acknowledgements

We thank M. Barkeshli, B. Doyon, P. Fendley, G. Giedke, J. Garcia-Ripoll, S. Iblisdir, A. Läuchli, T. Neupert, D. Poilblanc, A. Polkovnikov, K. Shtengel and S. Simon for inspiring conversations and useful comments. This work was supported by the EPSRC grants EP/I038683/1, EP/M50807X/1 and EP/P009409/1.

## Author contributions

All authors contributed to developing the ideas, analysing the results and writing the manuscript. C.J.T. implemented the algorithm.

## Additional information

**Competing interests:** The authors declare no competing financial interests.

**Publisher's note**: 

