## [Peer Review File · Nature Communications]

Reviewers' comments:

Reviewer #1 (Remarks to the Author):

In this paper authors propose a specific protocol for finding the optimal free model for many-body interacting systems. Their basic idea is interesting in that one doesn't assume that the creation/annihilation operators of the variational free model are linearly related to the operators of the microscopic model. I have the following questions for authors:

1. I am not sure, but it seems to me that in writing Eq.3 authors *assume* that a free model exists which commutes with the interacting density matrix ρ . Can authors point out why this is always the case?

2. Does the method work when ρ is the density matrix of the full system? (i.e. one doesn't trace out anything). If not, is the size of the subregion A over which ρ lives another variational parameter?

3. Consider the model in Eq. 5 with $h_z = 0$ i.e. classical Ising model in a magnetic field. Do authors find a free model with zero trace distance in this case? I wasn't able to tell this from Fig.4. Also, Fig.4 would be more useful (I think) if authors color-plot trace distance for the optimal model in the main figure, and not in the inset.

4. It seems that in authors' protocol whether the corresponding free model has bosonic or fermionic excitations is determined solely by the nature of the local Hilbert space, and is not an outcome of the variational procedure. For example, in the spin model studied in the paper, the local Hilbert space has two states, which is same as that for a spinless fermion. I can't see how this justifies authors' claim that the optimal model could have excitations which are allowed to be bosonic or fermionic, irrespective of the microscopic model. Does it even make sense mathematically to subtract a reduced density matrix of a spin-system from a reduced density matrix of a (free) bosonic system, which has different size of the Hilbert ($= \infty$). Note that hardcore bosons are not free and do not have any notion of single-particle entanglement energies.

5 As a tool, the method doesn't seem very useful in that one is restricted to small sizes due to the exponential (in system size) cost of finding ρ . Of course, the number of variational parameters scale linearly with N , but since one still needs the full ρ , the method is not scalable. Do authors have any approximate scheme where the method can be made to scale linearly with N and thus accesses relatively larger system sizes?

Reviewer #2 (Remarks to the Author):

The paper "Optimal free models for many-body interacting theories" by Turner et al. describes a procedure to optimally reproduce in a specific technical sense the behavior of an interacting quantum system using a free model. The work appears to be technically sound, and the results are interesting and at points surprising. In my opinion the most significant observation is that for the model considered, the free theory can exactly reproduce the interacting theory, at least in the sense of having zero trace distance between the approximate and exact theory.

The work will be interesting to some theoretical physicists, especially in condensed matter and AMO, who are interested in new tools to describe many-body systems. However, while it seems to reveal a new property of these systems --- that the trace distance of many-body states to a free model is zero everywhere in the phases of matter in their model -- I do not think it will find an unusually broad audience because the most interesting findings are specific to a single model

(moreover in 1D) and no strong arguments are given for their generality beyond this model, and there is no obvious route to turn the method into a tool that can approximate systems where we don't already have a more accurate answer, since it relies on knowing a priori the entanglement spectrum. Nevertheless, it may stimulate progress in such directions and therefore will be of interest to many-body theorists interested in pursuing these directions. The work is suitable for publication in a more specialized journal if some technical questions are addressed.

I have some technical questions regarding the findings:

1. The approach presented is different than mean field theory. However, there are similarities. The MFT is also an "optimal free model for many-body interacting theories." The MFT optimizes the energy over all free states, finding the free state with the closest energy to the ground state. The current paper instead optimizes the trace distance to the GS. The former approach has the advantage that it can be optimized without knowing the exact answer a priori, while the latter has the advantage that it (in some sense) optimally reproduces all physical observables, rather than just the energy. Would the authors agree with this assessment? I think that the relationship to and differences with MF deserve more discussion in the paper given the latter's central "tool of first resort" status for many theorists.

2. Related to this, there are statements such as "We find D_F to be small when MFT is applicable." Isn't this the definition of "MFT is applicable"? I.e. it reproduces the true states' observables (minimizes trace distance to the true state)?

3. pg 2, col 1. (This is likely a minor confusion from the wording) "As a consequence, the eigenstates of the optimal model σ are the same as the eigenstates of ρ ." I agree if you minimize over all models, but this can't be true if you are optimizing over free models can it? How can a free model reproduce an arbitrary state exactly? Does this mean that the only discrepancy between the true and approximate states is due to the weights on the eigenstates rather than the eigenstates themselves.

4. I have fairly fundamental technical confusion. I believe it is a confusion on my part rather than a defect of the paper's logic, but the presentation should be improved to clarify this. Namely, how is the definition of σ associated with free theories compatible with the structure of ρ of, say, the spin model? (Although the same question would arise for any model's ρ .) They seem to live in different Hilbert spaces. This is related to the comment that free states of both fermions and bosons are included in $\sigma \in F$ -- how can this be when the density operators for fermions and bosons don't live in the same Hilbert space? For example, in the finite system size case, the ρ and σ seem to be matrices with different dimensions unless the number of "sites" and "particles" of the free theory can be chosen with great flexibility. Even then it isn't clear that one can always choose these numbers to match the true theory's dimension.

5. The trace distance can be interpreted as the maximum over all projectors of measurement differences associated with those projectors in the states ρ and σ . Does $D_F \rightarrow 0$ signify then that all measurements exactly agree? Why would this be possible as $L \rightarrow \infty$ but not for finite L ?

6. For $D_F \rightarrow 0$ in the quantum phases, how relevant is the gapped nature of the phases?

7. I understand the statement that the free theory becomes exact for long-wavelength observables for irrelevant interactions. This is consistent with $D_F \rightarrow 0$ for $L \rightarrow \infty$, but I believe that the vanishing of D_F is a much stronger statement. In particular, D_F should be sensitive to more than long-wavelength observables. Do the authors agree? In this regard, the vanishing of D_F is indeed surprising, and this aspect should be emphasized in the discussion of relevant/irrelevant perturbations.

Reviewer #3 (Remarks to the Author):

The paper under review considers the problem of finding the closest quasifree model whose eigenvalue spectrum faithfully reproduces the entanglement spectrum of a given interacting model, be it a quantum spin system or otherwise. This search problem is carried out in the case of the Ising model and several quantitative results obtained.

This is, in my opinion, a novel question to investigate, and I am not aware of any similar approach in the literature. I am convinced that this interesting approach could lead to some intriguing further work.

I have, however, some crucial queries/concerns regarding the actual implementation and formulation of the idea in the submission.

Firstly, the search in (3) appears to be only well defined when the free models in question are fermionic, since otherwise there is no way to match up an infinite number of free boson eigenvalues with the (presumably) finite number of entanglement spectrum eigenvalues E_k ? Do I misunderstand something here, or is the approach limited to just free fermions?

Secondly, I am very puzzled by the Ising model results. The $h_x = 0$ model is exactly solved by the Jordan-Wigner transformation and we know that its eigenvalue spectrum is of free form. Indeed, thanks to the work of Peschel and Eisler we know that the entanglement spectrum is actually free (as is actually the case for other corner-transfer solvable models), even for the critical case. But the numerics seems to indicate that the distance to the nearest free model is large. I read through the numerical description several times and was rather confused about this: e.g, do the authors assert that $D_f > 0$ for the critical points or not?

Thirdly, I am missing a sanity check example: I think it is necessary to check the numerical routines against known free models themselves: do they find the free model? E.g., how bad are the local minima in the case where the entanglement spectrum has a free model?

Finally, the Ising model isn't particularly surprising (i.e., it basically confirms the physically obvious hypothesis that the model is basically free). I would prefer to see some results on a model expected to be not free, i.e., can the authors reject the hypothesis that the entanglement spectrum has a free model in an interesting example.

In summary, this is an interesting paper, but there are several crucial confusing parts which I'd definitely like to see cleared up before I could recommend it for publication.

Response to Reviewer 1

We thank the Reviewer for her/his careful assessment of our paper. The Reviewer recognises the innovative aspects of our work and suggests some clarifications on specific aspects of our paper before it can be recommended for publication in Nature Communications. She/He finds it interesting that the optimal free modes are in general non-linearly dependent on the original modes of the problem:

“Their basic idea is interesting in that one doesn’t assume that the creation/annihilation operators of the variational free model are linearly related to the operators of the microscopic model.”

The Reviewer has some specific suggestions. We fully took into account her/his comments, which helped us to improve the presentation of our work and extend our results. We now believe that our manuscript is suitable for publication in Nature Communications. Below is a detailed response to all of the points raised by the Reviewer.

1. *“I am not sure, but it seems to me that in writing Eq.3 authors assume that a free model exists which commutes with the interacting density matrix ρ . Can authors point out why this is always the case?”*

We thank the Reviewer for pointing out the need for further clarification regarding this part of our construction. In order to arrive to Eq. (3) from Eq. (1) we first rotate the matrix ρ so that it is diagonal. We then invoke the proof given in [Phys. Rev. A **77**, 042111(2008)] to justify that for any other matrix σ its distance from ρ is minimised when they commute and their spectra are ordered in the same manner. We then search directly for the spectrum of a σ ; this spectrum should be Gaussian and minimise the distance from the spectrum of ρ . This guarantees that the density matrices ρ and σ commute, and that σ is free. Note that in the basis where ρ is diagonal, a free commuting σ can be constructed by taking a diagonal matrix and making its elements $\{\lambda\}$ to correspond to free-fermion entanglement spectra as given in Eq. (2). To clarify this point we improved the derivation that leads to Eq. (2).

2. *“Does the method works when ρ is the density matrix of the full system? (i.e. one doesn’t trace out anything). If not, is the size of the subregion A over which ρ lives another variational parameter?”*

The method does indeed work if ρ is the full system’s density matrix, i.e., no partial trace is performed. In this case, the entanglement spectrum is trivial, $\{0, \infty, \infty, \dots\}$. In this case $D_{\mathcal{F}} = 0$ identically, signifying that the system as a whole is not interacting with anything else. In other words, $D_{\mathcal{F}}$ measures the degree of interactions of a partition with its complement.

The size of the subregion can indeed vary, as the Reviewer points out. To be more precise, we define a system as free if there exists another free system for which their corresponding reduced density matrices, ρ and σ , give $D_{\mathcal{F}} = 0$ for all partitions. Note that the partitions need to be large enough compared to the correlation length, ξ , of the interacting system. Otherwise, non-universal effects of the correlation length become apparent, as shown in Fig. 1 of this reply. A single partition larger than ξ is sufficient if the system is translation invariant in the bulk, which is true in our case.

3. *“Consider the model in Eq. 5 with $h_z = 0$ i.e. classical Ising model in a magnetic field. Do authors find that a free model with zero trace distance in this case? I wasn’t able to tell this from Fig.4.”*

Figure 1: The interaction distance in the FM Ising model as a function of the size of the partition $j = 1, \dots, L - 1$. The system size L is fixed ($L = 16$), and the curves correspond to several values of the longitudinal field h_x . Universal properties of $D_{\mathcal{F}}$ emerge when the entanglement partition is larger than the correlation length.

We thank the Reviewer for the question regarding the classical limit. For a classical state at $h_z = 0$, the entanglement spectrum contains a single level, thus the interaction distance is trivially zero. The numerical issue in Fig. 4 is due to an exponentially small degeneracy caused by the spontaneous breaking of the Néel symmetry. The finite-size splitting has a length scale set by h_z which vanishes on the classical line. Thus, close to the classical axis we do not find exact eigenstates because we cannot resolve the energy splitting without using this symmetry. Furthermore, at the classical critical point ($h_z = 0, h_x = 2$) there is an extensive degeneracy in the ground state.

Moreover, we thank the Reviewer for suggesting to modify Fig. 3 (previously Fig. 4):

Also, Fig.4 would be more useful (I think) if authors colour-plot trace distance for the optimal model in the main figure, and not in the inset.

We have followed this recommendation in the revised version of our manuscript. We have elucidated on the classical AF axis in the introduction of the “Application to the Ising model” paragraph and in the caption of Fig. 3.

4. “It seems that in authors’ protocol whether the corresponding free model has bosonic or fermionic excitations is determined solely by the nature of the local Hilbert space, and is not an outcome of the variational procedure. For example, in the spin model studied in the paper, the local Hilbert space has two states, which is same as that for a spinless fermion. I can’t see how this justifies authors’ claim that the optimal model could have excitations which are allowed to be bosonic or fermionic, irrespective of the microscopic model. Does it even make sense

mathematically to subtract a reduced density matrix of a spin-system from a reduced density matrix of a (free) bosonic system, which has different size of the Hilbert ($= \infty$). Note that hardcore bosons are not free and do not have any notion of single-particle entanglement energies.”

We thank the Reviewer for this important comment. In the new version of the manuscript, we have significantly clarified our presentation on how to address the apparent “mismatch” of Hilbert space dimension for different types of modes.

In particular, our procedure fixes in advance the statistics of the (free) degrees of freedom it uses. The statistics of the free modes thus is not a result of the variational procedure, but it determines how the free many-body entanglement spectrum is constructed from the variational single-body levels (Eq. (2) for the fermion case). Nevertheless, as the Reviewer points out, we must be more precise by what we mean when we compare a fermionic density matrix to another density matrix defined for a different kind of particles.

We emphasise that our method works with the spectra of ρ and σ , and that $D_{\mathcal{F}}$ is sensitive only to their largest eigenvalues. We elaborate on the issue of different Hilbert space dimensionalities referring to the singular value decomposition. A consequence is that these spectra can always be padded with zeros to make their dimensions match and Eq. (3) consistent. This is possible since this procedure does not change the rank of either reduced density matrices which is the physically relevant entity, contrary to their dimension. We append in the main text a paragraph titled “General optimal free description” that addresses this important point in detail.

Finally, we agree that soft-core bosons cannot arise in a system without interactions. However, in the present work we have used the term to refer the case where n_i in Eq. (2) is allowed to take integer values up to a certain cutoff. In order to avoid any confusion, we have modified the presentation of our method referring only to the fermionic case. We now mention the soft-core ansatz for the many-body levels in the Discussion/Conclusions section, presented as the simplest generalisation of the fermionic one in Eq. (2).

5. *“As a tool, the method doesn’t seem very useful in that one is restricted to small sizes due to the exponential (in system size) cost of finding ρ . Of course, the number of variational parameters scale linearly with N , but since one still needs the full ρ , the method is not scalable. Do authors’ have any approximate scheme where the method can be made to scale linearly with N and thus accesses relatively larger system sizes?”*

Our method *is* efficient in the computational sense. While indeed using exact methods to obtain ρ is exponentially expensive in the system size, there are numerous variational algorithms including Monte-Carlo routines and Tensor Network representations one can use to approximate ρ to sufficient accuracy. The computational cost of these methods scales polynomially with the system size. Tensor Networks are particularly suitable because they provide direct access to the entanglement spectrum, with the lowest entanglement energy being approximated with the high fidelity, to which $D_{\mathcal{F}}$ is most sensitive.

Finally, the calculation of $D_{\mathcal{F}}$ is itself a search problem. Its runtime complexity is polynomial in its input, i.e. the number, χ , of entanglement levels. This is performed efficiently by using well known minimisation methods as mentioned in the main text. Furthermore, considering that region A is a disk (reducing to a line interval in 1D) and symbolising the correlation length with ξ , we have the following for the runtime T . In 1D, for gapped states $\chi = \text{const}$ and for gapless states $\chi \sim L$, and thus typically $T \sim \text{poly}(\log L)$. In dD, we have for gapped states $\chi \sim \exp \xi^{D-1}$

due to the area law leading to $T \sim \text{poly}(\xi^{D-1})$, while gapless states have a logarithmic correction [Phys. Rev. D **92**, 126013 (2015)] so that $\chi \sim \exp \xi^{D-1} \log \xi$ giving $T \sim \text{poly}(\xi^{D-1} \log \xi)$.

In our particular case, the cost of finding ρ is *linear* in L . In order to demonstrate the numerical efficiency of our method, we have performed the variational minimisation using DMRG for large systems of up to 1024 spins. These results are consistent with the Exact Diagonalisation results of Fig. 2, and we have included them in the Methods section in the paragraph titled “Efficiency and convergence for DMRG and single-body levels”. We have also expanded the related discussion in the text in a new paragraph titled “Efficiency of computing $D_{\mathcal{F}}$ ”. In this paragraph we stress that our method requires only the ground state and in 1D its cost scales linearly with system size.

Response to Reviewer #2

We thank the Reviewer for her/his careful reading of our manuscript. The Reviewer finds our work technically sound and interesting; in addition, she/he characterises our results as surprising, pointing towards the novelty and usefulness of our approach:

“The work appears to be technically sound, and the results are interesting and at points surprising. In my opinion the most significant observation is that for the model considered, the free theory can exactly reproduce the interacting theory, at least in the sense of having zero trace distance between the approximate and exact theory. [...] The work will be interesting to some theoretical physicists, especially in condensed matter and AMO, who are interested in new tools to describe many-body systems.”

However, unlike the opinion of the other two Reviewers, she/he thinks that our work appeals only to a specialised audience because

the most interesting findings are specific to a single model and [...] it relies on knowing a priori the entanglement spectrum. Nevertheless, it may stimulate progress in such directions and therefore will be of interest to many-body theorists interested in pursuing these directions. The work is suitable for publication in a more specialized journal if some technical questions are addressed.

We respectfully disagree with the Reviewer on the potential scope of our paper. Apart from broad range of phenomena spanning condensed matter, AMO systems and quantum chemistry, many-body physics directly pervades many other fundamental areas including quantum statistical mechanics, quantum field theory and high-energy physics. In particular, the profound role of quantum entanglement which underpins these different subjects (from emergent ordered phases in condensed matter to black holes) is still a subject of active investigation. In this context, our work addresses a very basic yet fundamental question: when can the ground state of a quantum many-body system be viewed as composed of nearly-free degrees of freedom. We believe this question plays a fundamental role in entire theoretical physics, as most of our knowledge and intuition is based on free systems or simple solvable models.

More specifically, the Reviewer objected that our findings are “specific to a single model” and “rely on knowing a priori the entanglement spectrum”. We acknowledge that our method makes use of other techniques (such as exact diagonalisation) to obtain the entanglement spectrum. However, as we show in the revised version of our manuscript, obtaining the entanglement spectrum can be achieved efficiently for a large variety of models by other techniques such as tensor networks. We disagree with the Reviewer, however, that by getting the entanglement spectrum one has fully understood the physics of the given system. Indeed, the physics will only become clear once we identify what kind of (weakly correlated) emergent degrees of freedom produce such an entanglement spectrum. This crucial step is non-trivial, and our work makes progress on it by showing how the relevant degrees of freedom can be identified when they are nearly-free.

Finally, our results are not limited to a single model. To demonstrate this fact, we have included in the revised version of our manuscript a study of the strongly-interacting fermions in 1D in a new section with title “Maximally Interacting Model”. This analysis establishes a surprising connection with \mathbb{Z}_3 parafermions. Given these rich findings and the versatility and generality of our method, we thus disagree with the Reviewer that our paper is more appropriate for a specialised journal, and stand by our initial decision to submit it to Nature Communications.

1. *“The approach presented is different than mean field theory. However, there are similarities. The MFT is also an “optimal free model for many-body interacting theories.” The MFT optimizes the energy over all free states, finding the free state with the closest energy to the ground state. The current paper instead optimizes the trace distance to the GS. The former approach has the advantage that it can be optimized without knowing the exact answer a priori, while the latter has the advantage that it (in some sense) optimally reproduces all physical observables, rather than just the energy. Would the authors agree with this assessment? I think that the relationship to and differences with MF deserve more discussion in the paper given the latter’s central “tool of first resort” status for many theorists.”*

We thank the Reviewer for this comment. We partially agree with the Reviewer: mean-field theory (MFT) optimises the energy, however not over *all* possible free states. One of the central points of our work is that the optimal free degrees of freedom can be of different nature than the ones defined by the Hamiltonian, over which MFT self-consistently optimises. For example, MFT of interacting fermionic modes, is a free theory of linear combinations of the same modes. Our approach can identify “freeness” with respect to arbitrary fermionic modes that can be non-linear combinations of the original ones. Furthermore, as the Reviewer appreciates, our method is sensitive to the correlations in the ground state, and thus the optimal free state reproduces all physical observables with the error bounded by $D_{\mathcal{F}}$.

Importantly, our method is applicable to any entanglement spectrum, while MFT is applicable under certain conditions, such as weak correlations and spatial dimension higher than the upper critical dimension. In particular, the upper critical dimension for the Ising model is 4 [see, e.g., arXiv:1012.0653], and thus MFT fails to predict the critical exponents for dimensions 2 and 3. On the contrary with our method we manage to obtain them for 1D quantum Ising, which is equivalent to classical 2D Ising. Finally, a central difference between our method and MFT is their aim; the latter aims to solve the given Hamiltonian, whereas we are motivated by characterisation and classification of states. We have included this comment clarifying the connection with MFT in the Discussion section of the revised version of our manuscript.

2. *“Related to this, there are statements such as “We find $D_{\mathcal{F}}$ to be small when MFT is applicable.” Isn’t this the definition of “MFT is applicable”? I.e. it reproduces the true states’ observables (minimizes trace distance to the true state)?”*

We agree with the Reviewer that our wording here was not logically precise. In particular, *if* mean-field is applicable, *then* $D_{\mathcal{F}} \approx 0$. The converse is not necessarily true, due to the arguments given in the reply to point 1 above. We have reworded this point in the main text to avoid any confusion.

3. *“pg 2, col 1. (This is likely a minor confusion from the wording) “As a consequence, the eigenstates of the optimal model sigma are the same as the eigenstates of rho.” I agree if you minimize over all models, but this can’t be true if you are optimizing over free models can it? How can a free model reproduce an arbitrary state exactly? Does this mean that the only discrepancy between the true and approximate states is due to the weights on the eigenstates rather than the eigenstates themselves.”*

The eigenstates of the optimal free state have to be the same as those of ρ . This was proven in Ref. [Phys. Rev. A **77**, 042111(2008)]. Since we allow for any unitary transformation to diagonalise the reduced interacting state, the optimal free model may look interacting with respect to the original modes. It is only in this non-linearly related frame that the model appears free in terms of the new modes. That it is possible to find such modes is apparent from the spectrum

that has the combinatorial distribution of free particle energies. Hence, as the Reviewer points out, the only difference between the interacting state and the free state is in the weights of the eigenstates.

4. *“I have fairly fundamental technical confusion. I believe it is a confusion on my part rather than a defect of the paper’s logic, but the presentation should be improved to clarify this. Namely, how is the definition of σ associated with free theories compatible with the structure of ρ of, say, the spin model? (Although the same question would arise for any model’s ρ .) They seem to live in different Hilbert spaces. This is related to the comment that free states of both fermions and bosons are included in $\sigma \in \mathcal{F}$ – how can this be when the density operators for fermions and bosons don’t live in the same Hilbert space? For example, in the finite system size case, the ρ and σ seem to be matrices with different dimensions unless the number of “sites” and “particles” of the free theory can be chosen with great flexibility. Even then it isn’t clear that one can always choose these numbers to match the true theory’s dimension.”*

We thank the Reviewer for pointing out the need for better presentation of this important point. This reinforces our claim that the notion of freedom in many-body theory should be viewed in the context of the statistics of the potentially free modes. Before the optimisation is performed, we need to choose what statistics (Eq. (2)) the free degrees of freedom should obey. Regarding the free reduced state σ , it is fixed by our method’s setup to be the same size as the reduced interacting state ρ so that their Hilbert spaces are equal in size, regardless of what free manifold we chose to calculate the interaction distance from. We remind that since our method works with their spectra only and that $D_{\mathcal{F}}$ is sensitive only to their large eigenvalues, these spectra can always be padded with zeros to make their dimensions match and ensure Eq. (3) consistent. Importantly, this does not change the rank of either reduced density matrix, which is the physically relevant quantity.

To improve the presentation of our method, we develop its steps referring only to the fermionic case in Eq. (2). This removes any confusion about this point in the initial presentation of our measure $D_{\mathcal{F}}$. After that we elaborate in more detail on the possibility of using free quasiparticles of different statistics by adding a new paragraph. The presentation of how to resolve the apparent mismatch in Hilbert space dimensions is made in a new paragraph “General optimal free description” in terms of the singular value decomposition to make the argument precise.

5. *“The trace distance can be interpreted as the maximum over all projectors of measurement differences associated with those projectors in the states ρ and σ . Does $D_{\mathcal{F}} \rightarrow 0$ signify then that all measurements exactly agree? Why would this be possible as $L \rightarrow \infty$ but not for finite L ?”*

As $D_{\mathcal{F}}$ quantifies the error of measuring observables using σ instead of ρ , indeed $D_{\mathcal{F}} \rightarrow 0$ means that all measurements agree. This can happen either for finite L or for $L \rightarrow \infty$ depending on the model under consideration. In the example of the Ising model, this generally happens for $L \rightarrow \infty$ signifying that the model is asymptotically free. If we were to study a free model (such as the Ising model on the line $h_x = 0$) then $D_{\mathcal{F}} = 0$ for any size L of the system.

The interaction distance as we have defined it through a local cut of the system behaves as other known local order parameters. A common example is magnetisation, which can provide global information from local information. Furthermore, we have shown that $D_{\mathcal{F}}$ ’s contour lines follow the RG flow (Fig. 1) and that its scaling captures critical exponents (Fig. 2), establishing it as a physical quantity. Thus for finite sizes, we expect the interaction distance to be plagued by the same problems as any other physically relevant quantity. The relevant discussion is appended

in a new paragraph “Finite size scaling of $D_{\mathcal{F}}$ ”.

6. “For $DF \rightarrow 0$ in the quantum phases, how relevant is the gapped nature of the phases?”

This is an interesting question to which we do not have a general answer. Gapless systems are more delicate as the number of their entanglement spectrum levels is not bounded. We can however already see from Fig. 1 in the main text that, for a given system size, $D_{\mathcal{F}}$ has higher value near criticality where the correlation length ξ is large. We also demonstrate in the manuscript with a scaling analysis that these high- $D_{\mathcal{F}}$ regions shrink with system size.

Interestingly, if ρ is Gaussian (H is quadratic), then ξ does not induce such an effect near criticalities since now $D_{\mathcal{F}} = 0$ as H_E is also quadratic [J. Phys. A **36**, L205 (2003)]. Indeed, we note that the critical point on the ferromagnetic Ising free line ($h_x = 0, h_z = 1$) shows $D_{\mathcal{F}} = 0$ identically, for any system size.

More interestingly, we also find the antiferromagnetic Ising critical line (except the classical point at $h_z = 0$) to have $D_{\mathcal{F}}$ decaying with system size. Thus, we can say that at least some critical points are asymptotically free, based on the standard power-law scaling ansatz commonly employed in such cases (with critical exponents agreeing with the literature).

7. “I understand the statement that the free theory becomes exact for long-wavelength observables for irrelevant interactions. This is consistent with $D_{\mathcal{F}} \rightarrow 0$ for $L \rightarrow \infty$, but I believe that the vanishing of $D_{\mathcal{F}}$ is a much stronger statement. In particular, $D_{\mathcal{F}}$ should be sensitive to more than long-wavelength observables. Do the authors agree? In this regard, the vanishing of $D_{\mathcal{F}}$ is indeed surprising, and this aspect should be emphasized in the discussion of relevant/irrelevant perturbations.”

We thank the Reviewer for bringing up this very interesting point. We agree that the entanglement spectrum encompasses correlations on all length scales. As $D_{\mathcal{F}}$ is explicitly constructed from the entanglement spectrum, it also probes long-wavelength as well as short-distance physics.

We expect, as first pointed out by Li and Haldane [Phys. Rev. Lett. **101**, 010504 (2008)], that the entanglement spectrum generically separates into a long-wavelength part (i.e., low entanglement energies) which carries universal information about the system, and a short-distance part which is non-universal. The separation into long- and short-wavelength parts of the entanglement spectrum can also be directly seen in the example of a free fermion ladder [Journal of Statistical Mechanics: Theory and Experiment 2013, P08013 (2013)]. These different physical regimes can be probed by varying the linear size of the partition A relative to the size of quasiparticles, ℓ . Assuming that size of the partition A is much larger than ℓ , the long-wavelength information in the entanglement spectrum comes from correlated quasiparticle excitations across the entanglement partition. Our study focuses on this regime, where $D_{\mathcal{F}}$ displays universal properties which can be linked to the behaviour under renormalisation group and scaling (or CFT, in specific cases).

As shown in Fig. 1, as the subsystem A becomes small, $D_{\mathcal{F}}$ starts to probe the internal structure of the quasiparticles. However, we do not expect this short-distance information to yield any universal scaling, and thus we have not investigated this further.

Figure 1: The interaction distance in the FM Ising model as a function of the size of the partition $j = 1, \dots, L - 1$. The system size L is fixed ($L = 16$), and the curves correspond to several values of the longitudinal field h_x . When the partition size is large compared to ℓ , the size of the quasiparticles, $D_{\mathcal{F}}$ probes long-wavelength physics and obeys universal scaling; on the other hand, when the partition size is small, $D_{\mathcal{F}}$ effectively probes the short-distance (non universal) physics associated with internal structure of the quasiparticles.

Response to Reviewer #3

We thank the Reviewer for her/his appraisal of our manuscript and for finding it suitable for publication in Nature Communication, provided we clarify certain points. We were pleased to find the Reviewer recognises the novelty of the question we explore and its potential to shape the research landscape:

“This is, in my opinion, a novel question to investigate, and I am not aware of any similar approach in the literature. I am convinced that this interesting approach could lead to some intriguing further work.”

The Reviewer had several questions about our work which are addressed below.

“Firstly, the search in (3) appears to be only well defined when the free models in question are fermionic, since otherwise there is no way to match up an infinite number of free boson eigenvalues with the (presumably) finite number of entanglement spectrum eigenvalues E_k ? Do I misunderstand something here, or is the approach limited to just free fermions?”

We thank the Reviewer for pointing out the need for improvement in our presentation of the idea behind our work. The optimal free state σ is fixed by our method’s initial setup to be the same size as ρ , no matter what free ansatz we use for the entanglement levels in Eq. (2). Importantly, our method is *not* limited to fermions.

The Reviewer is correct, however, in that we need to be more precise about what we mean when we compare two density matrices defined for two different kinds of particles. To address this point we first give the general formalism for comparing systems comprising quasiparticles with different statistics and then we give a concrete example where this is successfully applied. In the revised version of our manuscript it is now explained that, since the ranks of the reduced states are the physically relevant quantities, we can pad their spectra with zeros (infinities in the entanglement spectrum) so that their dimensions match. We recall that our method works only with the spectra of ρ and σ , thus $D_{\mathcal{F}}$ is sensitive only to the large eigenvalues, and the spectra can always be padded with zeros to make their dimensions match and ensure the consistency of Eq. (3). To formulate precisely how we account for the apparent mismatch in Hilbert space dimensions we invoked the Schmidt decomposition. We have inserted these clarifications in a new paragraph named “General optimal free description”. We have also added new results in a new paragraph titled “Maximally Interacting Model” on a specific strongly-interacting 1D fermion system. In particular, we obtain a surprising result that the states of this system are described in terms of \mathbb{Z}_3 parafermionic degrees of freedom. This provides a concrete example of how we match the reduced states for two different kinds of particles.

Secondly, I am very puzzled by the Ising model results. The $h_x = 0$ model is exactly solved by the Jordan-Wigner transformation and we know that its eigenvalue spectrum is of free form. Indeed, thanks to the work of Peschel and Eisler we know that the entanglement spectrum is actually free (as is actually the case for other corner-transfer solvable models), even for the critical case. But the numerics seems to indicate that the distance to the nearest free model is large. I read through the numerical description several times and was rather confused about this: e.g, do the authors assert that $D_{\mathcal{F}} > 0$ for the critical points or not?

We thank the Reviewer for bringing up a possible source of confusion. Indeed, the FM Ising model is free on the entire line $h_x = 0$, as it maps to a free fermion Hamiltonian. Our results are fully consistent in that we find $D_{\mathcal{F}} = 0$ to numerical precision when $h_x = 0$, even at the

critical point.

We believe the confusion arises for the critical point in the FM case ($h_x = 0$ and $h_z = 1$). Here we find $D_{\mathcal{F}} = 0$ exactly (in agreement with Peschel and Eisler) but an infinitesimal perturbation $h_x > 0$ makes the model interacting, as shown in Fig. 2. Thus, the perturbation is strongly relevant in this case.

In the AFM case without interactions ($h_x = 0$), the situation is equivalent to the free FM case, and we again find $D_{\mathcal{F}} = 0$ to numerical precision. Upon turning on interactions ($h_x > 0$), the AFM model behaves very differently from the FM one, and remains critical along an entire line. Note that there are no exact results in the literature on this critical line, but numerical/perturbative study in [Phys. Rev. B **68**, 214406 (2003)] has argued that the universal properties along the critical line are the same as the free critical point ($h_x = 0$ and $h_z = 1$). Our results are consistent with this; we find $D_{\mathcal{F}}$ is in general non-zero for finite size L , but decays in thermodynamic limit (Fig. 2), and thus the entire critical line is approximately free.

Thirdly, I am missing a sanity check example: I think it is necessary to check the numerical routines against known free models themselves: do they find the free model? E.g., how bad are the local minima in the case where the entanglement spectrum has a free model?

The line $h_x = 0$ is free for any h_z , serving as a sanity check, and gives $D_{\mathcal{F}} = 0$ to numerical precision. This directly means that our algorithm reproduced perfectly the entanglement spectrum of the system (and thus any local or even non-local observables, like entropy, would agree to numerical precision).

Regarding optimisation, we implement the basin-hopping method to avoid overestimating $D_{\mathcal{F}}$ in case of local minima. However, a local minimisation in the case of the free FM/AFM Ising model is enough since the initial guess our algorithm constructs is good enough so that the global minimum is easily reached. In the revised version of our manuscript, we include a new section in the Methods where we provide new results obtained by DMRG. This extends our previous calculations on the Ising models to very large systems (1024 spins). With these results we confirm the scaling of $D_{\mathcal{F}}$ across much broader range of system sizes, recovering the expected scaling exponents (e.g., ν). This provides further evidence that our optimisation procedure in computing $D_{\mathcal{F}}$ does not get trapped in local minima, even in very large systems.

Finally, the Ising model isn't particularly surprising (i.e., it basically confirms the physically obvious hypothesis that the model is basically free). I would prefer to see some results on a model expected to be not free, i.e., can the authors reject the hypothesis that the entanglement spectrum has a free model in an interesting example.

We respectfully disagree with the Reviewer that the results on the Ising model are “not particularly surprising”. Once we add longitudinal field, h_x , as a perturbation, via the Jordan-Wigner transformation the Hamiltonian acquires a quartic term in the fermionic operators. In the ferromagnetic case, perturbed CFT approach can be applied in the vicinity of the free critical point. However, no exact results are available for strong values of the longitudinal field, nor in the AFM case where the model is non-integrable. Thus, we find it surprising that $D_{\mathcal{F}}$ with respect to free fermions decays with L for almost the whole parameter space, even with the strength of the interacting term being comparable with the kinetic terms.

Nevertheless, we have followed the recommendation of the Reviewer and investigated other mod-

els whose strong interactions cannot be simply expressed in terms of ordinary fermions in a new paragraph titled “Maximally Interacting Model”. We initially construct a model, given in terms of fermions, for which we can rigorously prove to have a finite value of $D_{\mathcal{F}}$ in the thermodynamic limit. Then, we investigated the physical meaning of such a state and discovered that it corresponds to a fixed point of a \mathbb{Z}_3 parafermion model. Thus, we obtain the surprising result that the “most interacting” fermionic states that can be constructed in 1D with Schmidt rank 4 have correlations that can be described in terms of simplest parafermion degrees of freedom. These models are in fact experimentally relevant [Ann. Rev. Cond. Mat. Phys. **7**, 119-139]. We believe the new section we have added on the \mathbb{Z}_3 model will satisfy the Reviewer’s request to study a genuinely interacting model in addition to the Ising model.

Finally, we would like to mention another example which we are currently investigating – the integer and fractional quantum Hall states. The integer case for filling factor $\nu = 1$ shows $D_{\mathcal{F}} = 0$ to numerical precision w.r.t. free fermions, as expected. The fractional cases $\nu = 1/2$ for bosons and $\nu = 1/3$ for fermions show a decaying $D_{\mathcal{F}}$ compatible with the composite fermion effective picture. These preliminary results are encouraging. However, a complete analysis of this topic lies beyond the scope of the current paper due to many immediate questions (e.g., the value of $D_{\mathcal{F}}$ for non-Abelian FQH states) and they are left for a future publication.

In summary, this is an interesting paper, but there are several crucial confusing parts which I’d definitely like to see cleared up before I could recommend it for publication.

We believe that we have addressed satisfactorily all the comments of the Reviewer. By performing major modifications to the paper and including new results beyond the Ising model, we hope that the Reviewer will find our manuscript suitable for publication in Nature Communications.

Detailed list of changes:

Main text:

1. We changed the title from “Optimal free models for many-body interacting theories” to “Optimal free descriptions of many-body theories” in order to emphasise the generality of our approach.
2. The presentation of our method is performed with respect to free fermions in the sake of clarity.
3. Introduced new paragraph “General Optimal Free Description” which contains generalisations beyond free fermions. The case of soft-core bosons is now mentioned in the Discussion section.
4. Introduced new paragraph “General Optimal Free Description” discussing the compatibility of entanglement Hilbert spaces, as requested by the Reviewers.
5. We have modified Fig. 3 (previously Fig. 4) as requested by Reviewer #2.
6. Inserted comments about the relation with mean-field theory in the introduction of the manuscript, in the introduction of the “Application to the Ising Model” paragraph, with the addition of the important distinction regarding the upper critical dimension in the Discussion section.
7. A new paragraph “Efficiency of computing $D_{\mathcal{F}}$ ” is added to elaborate on the runtime complexity of our method with the degree of entanglement, arguing that the computation polynomial in its input and thus efficient.
8. Introduced the paragraph “Finite size scaling of $D_{\mathcal{F}}$ ”, containing a discussion of long-wavelength vs. short-distance properties captured by $D_{\mathcal{F}}$.
9. We generalised our scaling ansatz in Eq. (4) to accommodate the fact that $D_{\mathcal{F}}$ is bounded, which is needed for our new DMRG results in large systems.
10. Added a new paragraph “Maximally interacting model”, extending our method to a genuinely interacting 1D system, which turns out to be connected with a parafermion chain.

Methods:

1. Move previous Fig. 3 to Methods (now Fig. 4).
2. Add new Fig. 5 which shows DMRG results for very large systems, confirming and extending our results in Fig. 1 and Fig. 2. This explicitly shows the scalability and efficiency of our method.
3. Added a paragraph containing an analytic proof of the maximisation of $D_{\mathcal{F}}$ for a rank-4 entanglement spectrum.

Reviewers' comments:

Reviewer #2 (Remarks to the Author):

The authors have carefully addressed my technical questions. The new paper, together with responses to myself and the other referees is much clearer to me. I believe that if some further technical questions are answered, the paper might be suitable for publication in Nature Communications.

Probably my biggest objection to publication was based on breadth of interest, and was around the criticism that it applying the optimal free theory method requires knowing a priori the entanglement spectrum. I believe that this objection largely still remains. (more on this below.) I believe this is what Referee 1 is also concerned with in his 5th comment, explaining that the method is scalable.

In particular, in order to apply the current method, one must have the entanglement spectrum. In some cases (though not all) this can be obtained efficiently, as the authors point out. However, in my mind the interestingness of the current method is somewhat limited: one understands things in a different way, but only for models that one can already solve efficiently. A much broader and very exciting potential application (that I believe Ref 1 also had in mind) is to build on the current paper to efficiently find optimal free Hamiltonians for models for which no other solution can be found efficiently. However, it is not at all how to make this leap. Therefore, the main advance here seems to be to allow one to understand observables in systems that one can already efficiently compute all of the observables in.

It is important for the authors to explicitly and clearly acknowledge this issue in the paper. However, once this limitation is explicitly acknowledged, I now agree the research presented in this paper is a worthy enough first step to in principle merit publication in such a high profile journal.

In their response, the referees completely clarified the comments 1, 2, 3, 4, and 6 (numbers referring to the labels in the authors' reply). I have some further questions for comments 5 and 7. In reverse order,

7. They authors say "Assuming that size of the partition A is much larger than ℓ , the long-wavelength information in the entanglement spectrum comes from correlated quasiparticle excitations across the entanglement partition." They then claim that by focusing on this regime, D_F displays universal properties, and imply that in this regime it is sensitive *only* to universal properties, not non-universal ones. However, I still don't understand this last statement. They justify this by stating that the entanglement spectrum generically separates into a long-wavelength part and a short-distance (non-universal) part. I agree completely.

But despite this, I am confused as to why the short-distance non-universal contribution to the entanglement spectrum (that should be present even as $\text{size}(A) \rightarrow \infty$) does not contribute to D_F .

5. I now understand better the $L \rightarrow \infty$ limit. But, for reasons related to the previous comment (#7) I'm still confused. Even if, for example, interactions in some basis are irrelevant perturbations (in RG sense), I don't see how this would lead to $D_F \rightarrow 0$ as $L \rightarrow \infty$. Again, why isn't D_F sensitive to the microscopic deviations? It seems like it should be since it measures deviations for *all possible* observables, including short-distance ones, even if $\text{size}(A)$ and L are large.

Reviewer #3 (Remarks to the Author):

I am satisfied with all the revisions and am happy to recommend acceptance.

Response to Reviewer 2

We thank the Reviewer for her/his detailed assessment of our manuscript, and for pointing out the need to clarify three final points.

Point 1:

“In particular, in order to apply the current method, one must have the entanglement spectrum. In some cases (though not all) this can be obtained efficiently, as the authors point out. However, in my mind the interestingness of the current method is somewhat limited: one understands things in a different way, but only for models that one can already solve efficiently. A much broader and very exciting potential application (that I believe Ref 1 also had in mind) is to build on the current paper to efficiently find optimal free Hamiltonians for models for which no other solution can be found efficiently. However, it is not at all how to make this leap. Therefore, the main advance here seems to be to allow one to understand observables in systems that one can already efficiently compute all of the observables in.”

“It is important for the authors to explicitly and clearly acknowledge this issue in the paper. However, once this limitation is explicitly acknowledged, I now agree the research presented in this paper is a worthy enough first step to in principle merit publication in such a high profile journal.”

Authors' Reply:

Our previous version of the manuscript explicitly stated that “our method is reliant on the efficiency of the current methods in the literature for computing the entanglement spectrum of the ground state”. Thus, our main goal was not to design a method to solve a many-body problem (in the sense of approximating the system’s ground state wave function), but rather to develop a new *diagnostic* tool based on the knowledge of the entanglement spectrum as an input. We have presented evidence that our method offers a new perspective on the physics which has remained hidden from any other method. To clarify this even further, we now emphasise in the Introduction that our method has a diagnostic character.

We agree with Reviewer 2, however, that our method inspires some exciting further applications, in particular on finding the free Hamiltonian which shares the same ground-state correlations in a systematic way. This lies beyond the current manuscript, but we mention it as a possible future direction in the Conclusions section.

Point 2:

7. They authors say “Assuming that size of the partition A is much larger than ℓ , the long-wavelength information in the entanglement spectrum comes from correlated quasiparticle excitations across the entanglement partition.” They then claim that by focusing on this regime, D_F displays universal properties, and imply that in this regime it is sensitive *only* to universal properties, not non-universal ones. However, I still don’t understand this last statement. They justify this by stating that the entanglement spectrum generically separates into a long-wavelength part and a short-distance (non-universal) part. I agree completely.

But despite this, I am confused as to why the short-distance non-universal contribution to the entanglement spectrum (that should be present even as $\text{size}(A) \rightarrow \infty$) does not contribute to D_F .

Authors' Reply:

We thank the Reviewer for requesting further clarification of this subtle point. The separation into a universal and non-universal part, as introduced by Li and Haldane, applies primarily to the case of gapped (topological) states. In this case, there is an exponentially large gap separat-

ing the two parts of the spectrum (as in, e.g, our Eq. (3)). Thus, the non-universal part, which is associated with the internal structure of the quasiparticles, is expected to be an exponentially small correction to $D_{\mathcal{F}}$. Moreover, this is a boundary effect of the virtual edge created by the entanglement partition, thus it is not expected to increase with subsystem size.

In gapless states, the non-universal part may not scale according to our Eq. (4) but could, in principle, still contribute to $D_{\mathcal{F}}$. Furthermore, in general there may not be a clear-cut separation of the entanglement spectrum into a universal and non-universal part. As we explicitly state in the text, we focus on the cases where the quasiparticles of the system are well-defined. In such cases, we still expect the dominant contribution to $D_{\mathcal{F}}$ from the universal part of the entanglement spectrum. In 1D, this is certainly true because the entire entanglement spectrum at scale-invariant critical points is universal and determined by the central charge of the associated conformal field theory [arXiv:0806.3059] (in higher dimensions, properties of the entanglement spectrum are subject of on-going work [arXiv:1603.02684, arXiv:1605.03629]).

Point 3:

5. I now understand better the $L \rightarrow \infty$ limit. But, for reasons related to the previous comment (#7) I'm still confused. Even if, for example, interactions in some basis are irrelevant perturbations (in RG sense), I don't see how this would lead to $D_{\mathcal{F}} \rightarrow 0$ as $L \rightarrow \infty$. Again, why isn't $D_{\mathcal{F}}$ sensitive to the microscopic deviations? It seems like it should be since it measures deviations for *all possible* observables, including short-distance ones, even if size(A) and L are large.

Authors' Reply:

We agree with the Reviewer that the behaviour of $D_{\mathcal{F}}$ in the thermodynamic limit is not completely analogous to that of a local observable under RG flow. The scaling exponent ζ tells us about the effect of interactions as the system size is scaled to infinity, and if $\zeta > 0$ the interactions are 'irrelevant'. However, this does *not* preclude a possibility that interactions are irrelevant in the usual RG sense, yet $D_{\mathcal{F}}$ remains finite as $L \rightarrow \infty$. Indeed, this is one of the reasons we included the parameter θ in our scaling ansatz, Eq. (4). The non-universal short-distance contributions to $D_{\mathcal{F}}$ manifest as a finite value for θ (see the fit in Fig. 5).

We have included clarifications of the Points 2 and 3 in the text surrounding Eq. (4).

Detailed list of changes:

1. Emphasised “diagnostic” character of the method in Introduction:

“The interaction distance is a diagnostic tool which provides information about the renormalisation properties of the model.”

2. Clarified the scaling of $D_{\mathcal{F}}$ and relation to RG properties around Eq. (4):

Assuming that the linear size of the systems quasiparticles, ℓ , is much smaller than the linear size of the partition A , the long-wavelength information corresponds to correlated quasiparticle excitations across the entanglement partition, while the short-distance physics is associated with internal structure of the quasiparticles. The non-universal part is thus a boundary effect which is insensitive to variation in the subsystem size. In the thermodynamic limit, the non-universal part is exponentially suppressed in a gapped phase [6], as seen from (3), and $D_{\mathcal{F}}$ then predominantly describes the universal properties of the system.”

and

“... the effect of interactions in the renormalisation group sense. For example, when interactions are relevant, $D_{\mathcal{F}}$ should remain non-zero as L increases, which dictates $\zeta \leq 0$. On the other hand, when interactions are irrelevant, it is expected that $D_{\mathcal{F}}$ decreases with L near the critical regions, in which case $\zeta > 0$. Note, however, that it is possible for interactions to be irrelevant and still yield finite $D_{\mathcal{F}}$ in the thermodynamic limit. This is because, $D_{\mathcal{F}}$ may be sensitive to non-universal (short distance) properties of the system, which can give a residual non-zero contribution parametrised in Eq. (4) by θ .”

3. Added a possible future application of the method in Conclusions:

“Further, our diagnostic $D_{\mathcal{F}}$ shows that the ground state of this interacting model in its gapped phases is well described by a free state, requiring exponentially less information to represent than the exact ground state. We expect that this could be generalised to an efficient algorithm for finding a representative (nearly) free state, similar to the DMRG method which constructs a low Schmidt rank approximation to gapped 1D ground states.”

REVIEWERS' COMMENTS:

Reviewer #2 (Remarks to the Author):

I am satisfied with the authors latest revisions. It is suitable for publication.